# Hybrid Bio-Inspired Structure Based on Nacre and Woodpecker Beak for Enhanced Mechanical Performance

**DOI:** 10.3390/polym13213681

**Published:** 2021-10-26

**Authors:** Zhongqiu Ding, Ben Wang, Hong Xiao, Yugang Duan

**Affiliations:** State Key Laboratory for Manufacturing Systems Engineering, School of Mechanical Engineering, Xi’an Jiaotong University, Xi’an 710049, China; dingzhq@stu.xjtu.edu.cn (Z.D.); wangben@xjtu.edu.cn (B.W.)

**Keywords:** hybrid bio-inspired design, nacre, woodpecker beak, mechanical performance, failure mode, finite element simulation

## Abstract

Materials with high strength and toughness have always been pursued by academic and industrial communities. This work presented a novel hybrid brick-and-mortar-like structure by introducing the wavy structure of the woodpecker beak for enhanced mechanical performance. The effects of tablet waviness and tablet wave number on the mechanical performance of the bio-inspired composites were analyzed. Compared with nacre-like composites with a flat tablet, the strength, stiffness and toughness of the novel hybrid nacre-like composite with tablet wave surface increased by up to 191.3%, 46.6% and 811.0%, respectively. The novel failure mode combining soft phase failure and tablet fracture revealed the key to the high toughness of composites. Finite element simulations were conducted to further explore the deformation and stress distribution of the hybrid brick-and-mortar-like structure. It showed that the hybrid brick-and-mortar-like structure can achieve a much better load transfer, which leads to greater tensile deformation in tablet before fracture, thus improving strength and energy absorption. These investigations have implications in the design of composites with high mechanical performance for aerospace, automobile and other manufacturing industries.

## 1. Introduction

Many natural materials have an unusual combination of stiffness, low weight, strength and toughness beyond the reach of current engineering materials. The mechanical properties of natural materials far exceed the components that make them up. It is mainly due to the well-organized microstructure and abundant effective interface interactions on multiple length scales [1,2,3,4], which provides infinite inspiration for the manufacture of new biomimetic structural materials. Nacre is an excellent example of such materials, which is mostly made up of crystalline aragonite (CaCO_3_) platelets (95% vol.) and bonded by a thin layer of biopolymer (5% vol.) [5,6,7,8]. Despite the high mineral content, nacre is almost 20–30 times tougher than aragonite alone [9,10]. The impressive mechanical property of nacre can be attributed to its brick-and-mortar structure [11,12], as shown in Figure 1a. In addition, the detailed sub-level structures, such as mineral bridges found in the organic matrix layers [13], nanoscale mineral islands found on the top and bottom surface of tablets [8] and tablet interlocks [7,14], also contribute to the toughness of the nacre. Many researchers have studied the effects of structure details on the mechanical performance of the composites inspired by nacre. Ghimire A et al. have designed nacre-like composites with interlocked tablets and analyzed the effects of waviness on mechanical properties of nacreous composites; results showed that increasing the tablet waviness can improve the stiffness, strength and toughness of the nacre-like composites [15,16,17,18]. Mirzaeifar et al. demonstrated that the existence of a hierarchical architecture in the designing of brick-and-mortar-like structures leads to superior defect-tolerant and structural properties [19,20]. Gu et al. systematically elucidate the effects of the density of mineral bridges on the mechanical response of nacre-inspired additive manufactured composites [21]. Jabir et al. [22] studied the influence of the material compliance gradient in mortar of nacre-like composites and proved the significant contribution of material compliance gradient to the mechanical performance. Although many researchers have studied the effects of brick-and-mortar structure and substructures on the mechanical performance which has provided guidance for the design of tough bio-inspired materials, they all revolve around the microstructure observed in the nacre itself.

However, only mimicking the brick-and-mortar structure and substructure of nacre makes a limited contribution to further enhancing the performance of such materials, because their properties are adapted to specific living environment. Actual service scenarios may have more complex and demanding performance requirements for engineering materials. Multibiological multiscale biomimetic design may be a promising design approach. Thus, we turn attention to other high-performance biomaterials, seeking inspiration for the design of high-performance composites with multiple strengthening and toughening mechanisms. Recently, the woodpecker beak has aroused the interest of scientists owing to its ability to withstand high impact [23]. In nature, a woodpecker beak repeatedly strikes into a tree trunk at a speed of 6–7 m/s, with an impact deceleration of 1000 g, without any recorded damage to the beak or brain [24,25]. Lee et al. suggested that the tightly packed keratin scales with wavy surface organized in an overlapping arrangement play an important role in resisting fracture during high-speed pecking [26], as shown in Figure 1b. Although the beak of other birds also shows this wavy structure, the waviness of those birds’ beaks is smaller than that of the woodpecker beak [27]. This further highlights the role of the wavy surface structure in tuning the mechanical performance to suit biological functions. Ha et al. proposed a novel bio-inspired honeycomb sandwich panel based on the microstructure of the wavy structure in the woodpecker beak, which indeed exhibits superior energy absorption capability compared with the conventional honeycomb sandwich panel [28].

In this work, we aim to integrate different toughening strategies of the nacre and woodpecker beak to achieve higher mechanical performance amplification in the given material. A novel hybrid brick-and-mortar-like structure with wavy surface tablets was proposed. The multi-material 3D-printing technique allows us to exercise complete control over the tablet structure design. The influences of the tablet waviness and tablet wave number on the mechanical response of the brick-and-mortar-like structure and their behavior mechanism were studied experimentally and numerically. The results in this study can be used in the design of advanced tough composites.

## 2. Materials and Methods

### 2.1. Design of Hybrid Bio-Inspired Structures

In this study, the proposed brick-and-mortar-like structure combines the brick-and-mortar structure of the nacre with the wavy structure of the woodpecker beak, including discrete hard blocks bonded by soft interfaces, as shown in Figure 2. The height of the tablet *h* is 1.5 mm, the length of the tablet *l* is 7.5 mm, the overlapping length of the tablet is designed to be half the length of the tablet and the width of the tablet *w* is 3.14 mm. The horizontal surfaces of the tablet are sine wave-like interfaces with wavelength *λ* and amplitude *A*. To quantify the waviness of wavy surfaces in tablets, a non-dimensional geometric parameter *wv* is defined as *wv* = *A/λ*. Since the length of the tablet *l* is fixed, the wavelength *λ* is controlled by the wave number *n*. In order to investigate how the horizontal wavy surfaces of the tablet impact the mechanical performance of the composites, five waviness (0, 0.3, 1, 2 and 3) and five wave numbers (6, 7, 8, 9 and 10) were considered. The nomenclature for designs is NaWb, where N is the wave number with a being its value, and W means the waviness with b being its value. For example, N8W03 means that the wave number is 8 and the waviness is 0.3; N8W10 means wave number is 8 and the waviness is 1. For all designs, the in-plane thickness *t* of the soft interfacial layer fixed as 0.3 mm in consideration of the 3D printer limitations. However, the stiff phase volume *f_v_* of the composites are not the same. More detailed dimensions of each design are listed in Table 1.

The relevant geometries were generated using Solidworks (Dassault Systèmes SolidWorks Corporation, Waltham, MA, USA). The 3D models were created by extruding the 2D designs and rendering them into Stereolithography (.stl) files. In addition, the test models consisting of the above-mentioned hard blocks were designed with dog-bone-like ends in order to follow ASTM (American Society of Testing Materials) standards, giving them appropriate dimensions suitable for tensile testing, as shown in Figure 2.

### 2.2. Sample Fabrication

All specimens used in the study were fabricated using a Stratasys J750 multi-material 3D printer (Stratasys, Minneapolis, MN, USA), which makes complex geometry with around 100 μm printing resolution [29]. Two of Stratasys’ commercial photopolymers, VeroWhite and TangoPlus, with strongly contrasting material properties, were used for the composites manufacturing a single print. VeroWhite is a white, stiff/rigid polymer representing hard tablets, and TangoPlus is a rubber-like transparent polymer in place of a biopolymer interface. Using Stratasys’ technology, the two materials are sprayed simultaneously as liquid layers and then cured in situ by UV light. This instant curing ensures perfect interfacial adhesion between the two different materials [21]. At the same time, the intermixing of different liquid polymers before curing creates an interface between the two materials, resulting in the mechanical properties of the printed composite depending on the printing direction [21,30]. Thus, all specimens were printed along the same orientation to avoid the influence of the layer orientation on the mechanical properties of the specimens. Figure 3a shows images of a representative 3D-printed specimen. After printing, the water jet was used to remove the gel-like support material from the samples. For the saturation of the curing, the as fabricated specimens were kept at room temperature for 24 h before mechanical testing.

### 2.3. Mechanical Testing

To capture the mechanical response of the bio-inspired composites, quasi-static uniaxial tensile tests were performed using an universal testing machine (MTS Systems Corp, Minneapolis, MN, USA) endowed with a 25 kN load cell. The specimens were clamped in place using serrated steel grip faces attached to steel vice action grips. We employed a slow displacement rate 0.2 mm/min to overcome the viscoelastic effect on the mechanical properties, because the soft phase in our composite specimen is highly stretchable before fracture. The test proceeded until the crack propagates thoroughly through the specimen and the load dropped. Strain gages were used to measure the strains on the samples. Three specimens were fabricated for each type of design. Load–displacement curves from tensile tests were transferred to nominal stress–strain curves, while stiffness and strength were calculated based on these curves. The toughness here was defined as the area under the stress–strain curve.

### 2.4. Finite Element Analysis

Numerical simulations were conducted in ABAQUS (ABAQUS Explicit, version 2017, ABAQUS Inc., Providence, RI, USA) to study the mechanical response of the composites. The two constituent phases used in printing were considered as isotropic materials. For VeroWhite, a power law plasticity model was used to model the initial yield and hardening, and a linear damage evolution law defined by final fracture strain was used to capture the softening. A linear plastic hardening model with linear damage evolution was used to model the stress–strain behavior of TangoPlus. The detailed mechanical properties implemented in the ABAQUS software are presented in Table 2. Studies show no interfacial debonding occurs in the composites fabricated by 3D-printing, due to the perfect adhesion between two phases obtained from the in situ UV curing [31]. Thus, a ‘tie’ constraint was used for the connection between the tablets and the soft interfacial layer. General contact with a ‘hard contact’ relationship was used to prevent the penetration of the contact pairs into each other. All models were generated by 3D stress elements with reduced integration C3D8R and meshed after a convergence test. Displacement boundary conditions were applied in the loading direction to simulate the experimental conditions. The left side of the model was held fixed, and the right side was stretched.

## 3. Results and Discussions

### 3.1. Experimental Tensile Test Response of Bio-Inspired Composites

Tensile tests were performed on all the presented bio-inspired composites. The results showed good repeatability in terms of stress–strain response and failure modes. For the sake of brevity, only one representative stress–strain curve and failure mode were reported for each sample series.

#### 3.1.1. Influence of Tablet Waviness on Mechanical Behavior

Figure 4a shows the stress–strain curves from the tensile test results of bio-inspired composites with various tablet waviness, where the feature points were marked. Although all the specimens show uniform deformation at the initial tensile stage, the fracture behaviors of different specimens are obviously different and can be divided into three failure modes, as shown in Figure 4b. The first peak stands for the strength of vertical short interfaces, which is defined as *σ_I_*; the last peak stands for the strength of horizontal interfaces, which is defined as *σ_II_*. The mechanical behaviors of composites with the tablet waviness below 2 match Mode I: a two-stage fracture with low failure stress. Upon loading, the vertical short interface fractures first, leading to a softening stage in the stress–strain curve followed by a hardening stage; the shear deformation of soft material along the horizontal interface takes place, soon after which the composites rupture completely. Such two-stage fracture behavior is also reported for 3D-printed composites inspired by interlocks [16] and mineral bridges [19] of nacre and bone [31]. The mechanical behavior of composites with a tablet waviness of 2 meets Mode II: a three-stage fracture with high failure stress. Besides the softening and hardening stage, a tablet deformation stage appears due to the tensile deformation in the tablets. In addition, the mechanical behavior of composite with tablet waviness above 2 matches Mode III: a single-stage fracture with relatively high failure stress, which mainly corresponds to the tablet deformation. The corresponding deformation and stress distribution of these three different failure stages are shown in Figure 4c. These phenomena indicate that adjusting the tablet waviness can change the failure mechanisms of composites.

Figure 4d compares the *σ_I_* and *σ_II_* of composites with tablet waviness below 3. It can be seen that both the *σ_I_* and *σ_II_* show an increasing trend with the increase in tablet waviness. The *σ_II_* increases significantly, from 1.3 MPa to 6.2 MPa, while the *σ_I_* increases from 2.1 MPa to 3.8 MPa for the composites in the order N8W00, N8W03, N8W10 and N8W20. Notably, the value of *σ_II_* for composite N8W10 firstly exceeds that of its *σ_I_*. Moreover, the strain at the beginning of the softening stage also tends to increase, as shown in Figure 4a, which indicates that a larger tablet waviness is beneficial to the delay of crack generation and the energy absorption. Meanwhile, it can be seen from the strain–stress curves that as the tablet waviness increases, the hardening in composites with tablet waviness below 3 changes from a non-linear and unclear increase to a linear and obvious increase.

To further analyze the hardening behavior in relation to the tablet waviness precisely, we quantify three critical aspects of the stress–strain response, which are the strain period during the hardening stage (*ε_h_*), the stress increased during the hardening stage (*ε_h_*) and the energy absorption (*U_h_*) during the hardening stage (the area under the stress–strain curve at hardening stage), as shown in Figure 4b. It can be found from Figure 4e that the values of *σ_h_*, *ε_h_* and *U_h_* exhibit a strong dependence on the tablet waviness. Flat interfaces in N8W00 result in the lowest stress increase (*σ_h_* = 0.13 MPa), whereas larger tablet waviness leads to a significant increase in the value of *σ_h_* (*σ_h_* = 0.29 MPa in N8W03, *σ_h_* = 1.55 MPa in N8W10 and *σ_h_* = 2.48 MPa in N8W20). This indicates that composites with larger tablet waviness can withstand higher stress and exhibit larger resistance to shear deformation resulting from tablet sliding. Compared with the *ε_h_* value of composite N8W00 (*ε_h_* = 0.016), the *ε_h_* value of composite N8W03, N8W10 and N8W20 increases by 139.3% (*ε_h_* = 0.039), 203.5% (*ε_h_* = 0.047) and 262.5% (*ε_h_* = 0.051), respectively. The composite N8W20 undergoes the largest hardening period and exhibits the highest hardening rate. In addition, the energy absorption during the hardening stage *U_h_* displays an increasing trend upon the increase in the tablet waviness from 0 to 2. From the above results, it is clear that the rise in increased stress during the hardening stage of the composite with larger tablet waviness supports its increment in tablet deformation, leading to the increment of strain whilst hardening, thereby promoting the energy absorption. To further analyze the influence of tablet waviness on the deformation of composites, we quantify two critical aspects: the strain period during the tablet deformation (*ε_p_*) (Figure 4b) and the energy absorption (*U_p_*) during the tablet deformation (the area under the stress–strain curve at tablet deformation stage). It can be seen that both the *ε_p_* and *U_p_* decrease when the tablet waviness increases from 2 to 3, causing earlier rupture. This is because the hardening in the hardening stage of composite N8W20 is not complete and continues to the subsequent deformation stage. In other words, the deformation stage of composite N8W20 is the result of the combined effect of horizontal interface shear deformation and tablet deformation, while the deformation of composite N8W30 only comes from tablet deformation.

Figure 5a shows the normalized surface area (normalized by the surface area of composite N8W00) and the scale factor (defined as *λ*/*w*) of composites for various tablet waviness, Figure 5b–d show the strength, stiffness and toughness values of composites with different waviness. Obviously, the stiffness increases as the tablet waviness increases, which can be attributed to the increase in contact interface. Compared with the stiffness of composite N8W00 (159.09 MPa), the stiffness of composite N8W03, N8W10, N8W20 and N8W30 increases by 10.6% (175.98 MPa), 25.2% (199.14 MPa), 46.6% (233.17 MPa) and 69.4% (269.55 MPa), respectively. However, there is indeed an optimal structure design based on the geometry and mechanical properties of the constituents under the circumstance of strength and toughness. Among all composites, composite N8W20 has the highest strength (6.2 MPa) and toughness (562.8 KJ/m^3^), which are 191.3% and 811.0% higher than that of composite N8W00. Interestingly, the increase in toughness of our nacre-like composite is obviously higher than that of other reported nacre-like composites [15,16,17,18,19,21]. Although the strength of composite N8W30 is slightly lower than that of composite N8W20, there is a sharp drop in the toughness of composite N8W30 compared with composite N8W20 due to smaller failure strain. This is essentially attributed the competition between interface hardening and stress concentration, which was also discussed by Horacio D et al. [16,18]. It should be noticed that the stiff phase volume in composites is not the same; composite N8W20 exhibits larger strength and toughness, although the stiff phase volume for composite N8W30 is higher than composite N8W20, which can be attributed to the wavy interface design.

#### 3.1.2. Influence of Tablet Wave Number on Mechanical Behavior

Figure 6a show the stress–strain curves from the tensile test results of composites with various wave number. It can be seen that these composites fail in a similar way, and their mechanical behaviors meet Mode II: a three-stage fracture with high failure stress. On the whole, although the tablet wave number does not have much effect on the mechanical behavior of composites compared with the tablet waviness, some regularities can still be observed. Figure 6c shows the *σ_I_* and *σ_II_* of composites with different tablet wave numbers. It is clear that the *σ_I_* of composite N6W20, N7W20, N8W20, N9W20 and N10W20 are basically constant, while the *σ_II_* of these composites have a slight decreasing trend as the tablet wave number increases. Figure 6d compares the increased stress during hardening *σ_h_* and the strain period during hardening *ε_h_*. When the tablet wave number increases to 10, there is an effective reduction in the value of *σ_h_*, which indicates the composite N10W20 exhibits the lowest hardening rate. We also find that the *ε_h_* increases first and then decreases as the tablet wave number increases, the composite N8W20 exhibits the largest *ε_h_* (0.051). Additionally, the energy absorption during hardening *U_h_* plotted in Figure 6e displays a similar changing trend with the increase in wave number. It is clear from Figure 6e that the tablet wave number has a greater influence on the deformation stage than the hardening stage. Both the strain period during deformation *ε_p_* and energy absorption during deformation *U_p_* display an increasing trend upon the increase in the tablet wave number. Compared with the composite N6W20, the *ε_p_* and *U_p_* of composites with other tablet wave numbers increase by up to 223.7% and 170.8%, respectively. To further analyze the influence of tablet wave number on the deformation of composites, we quantify another aspect of stress–strain response, the equivalent stiffness during the deformation stage (*E_p_*) as shown in Figure 6b, to describe the deformation resistibility capacity of the structure. As discussed previously in failure mode II, the deformation stage of these composites is a combination of horizontal interface shear deformation and tablet deformation. Figure 6f evinces that the *E_p_* of composites increases as the tablet wave number increases. This is likely due to the different degree of shear deformation of the horizontal interface and different stress in tablet at the deformation stage. As the deformation stage is the extension of hardening, the greater *ε_h_* and *σ_II_* means more adequate hardening, that is, larger horizontal interface shear deformation and greater stress in tablet. Thus, it inevitably leads to earlier failure of the horizontal interface and tablet at the deformation stage. The above discussion leads to a conclusion that although the increase in tablet wave number will reduce the load transfer, it will delay the failure of composites. In contrast, the decrease in tablet wave number will promote the load transfer but lead to premature failure of the composite.

Figure 7a shows the normalized surface area (normalized by the surface area of composite N6W20) and the scale factor (defined as *λ*/*w*) of composites for various wave number. Figure 7b–d show the strength, stiffness and toughness values of composites with different wave number. Although the contact interface area is the same, the mechanical properties are different due to the scale effect of the of microstructure design in nacreous composites. This result agrees with [32], where scale effect was studied for additively manufactured two-phase composites. It is clear that the strength displays a slight decreasing tend as the wave number increases. Compared with the strength of composite N6W20 (6.33 MPa), the strength of composite N7W20, N8W20, N9W20 and N10W20 reduces by 1.6% (6.23 MPa), 2.1% (6.20 MPa), 5.1% (6.01 MPa), 12.6% (5.54 MPa), respectively. In addition, the stiffness decreases as the tablet wave number increases, while there is indeed an optimum toughness. Among all the composites with same waviness, the composite N8W20 exhibits the greatest toughness, suggesting that composite N8W20 exhibits the optimal balance between interface hardening and stress concentration caused by geometrical scale.

The above results indicate that both the tablet waviness and tablet wave number can affect the mechanical response of the bio-inspired composite proposed in this work. By contrast, the tablet waviness has a greater impact on the composite than the tablet wave number. It is manifested in two aspects. On the one hand, the value of strength, stiffness and toughness of the composite changes more greatly due to the change of the tablet waviness. On the other hand, the transformation of failure mechanisms of the composite can be realized by tuning the tablet waviness, whereas tuning the tablet wave number cannot. The wavy interface design of the nacre-like composite offers additional resistance to shear effectively at macro scales, resembling the wavy surface of scales in woodpecker beaks, can boost the interface hardening and delay the fracture, leading to enhanced energy absorption.

### 3.2. Fracture Mechanisms and Morphologies

The typical failure patterns of the bio-inspired composites are shown in Figure 8a. The fracture patterns of the composites can obviously be divided into three types: the soft phase (interface) failure, the soft phase failure coupled with the tablet break and the tablet break. In composite N8W00, N8W03 and N8W10, soft phase failure in both the vertical and horizontal interface is the cause of the composite failure. Additionally, their zig-zag fracture path just proves that the shear deformation of soft phase during tablet sliding is the main mechanism of their energy absorption. Though similar fractures existed in composite N8W00, N8W03 and N8W10, we observed from Figure 8b that increasing the tablet waviness results in the increase in the fracture region area. Additionally, we quantified the fracture behavior of composites with tablet waviness below 2, as shown in Figure 8c. The increase in tablet waviness from 0 to 1 leads to an increase in the fracture region area up to two times. As discussed in Section 3.1.1, increasing the tablet waviness can improve the resistance of the composites to crack initiation and propagation. It can be seen from Figure 4a that the strain corresponding to the strength *σ_I_* increases with the increase in tablet waviness, so the deformations spread to a larger area and cause tablet pullout in a larger region, as previous studies reported [15], leading to an increase in the fracture region of composite N8W00, N8W03 and N8W10.

Figure 8 W20 and other composites with the same tablet waviness, the overall failure mechanism changes: soft phase failure and tablet break act synergistically leading to the special fracture morphology, that is, the tablets are completely broken, while the fracture path still displays a small zig-zag tooth pattern. This partial failure of the soft interface is due to the limited tablet sliding during tensile deformation. In spite of this, the incomplete failure of soft phase promotes the delay of fracture and the increase in energy absorption. Meanwhile, a higher portion of load is transferred to the tablet due to the increased interface area, which is beneficial to improve the load bearing capacity of composites. However, the interfacial hardening strength increases as the strain increases and when it is greater than that of the tablets themselves, localized tablet break was observed. As the tablet waviness continues to increase, localized tablet break prevails and the function of soft phase failure declines in the failure of composites. This has been proven by the fracture pattern of composite N8W30. It can be seen that the fracture is almost neat, with obvious brittle fracture characteristics. The horizontal interface area in composite N8W30 is large enough that a higher portion of the load is transferred to the tablets, thus the shear deformation in soft phase is averted; at the same time, the narrower top of the wavier interface promotes the generation of stress concentration zones, which causes the tablet crack to expand rapidly leading to fracture. Although the tablets break in composite N8W20 and N8W30, the results in Figure 4f show that the strain period during tablet deformation stage *ε_p_* of N8W20 is larger than that of N8W30. This can be correlated to the decrease in the strain during the hardening *ε_h_* of N8W20 (Figure 4e); the horizontal interface of N8W20 does not completely fail in the hardening stage and continues in the tablet deformation stage, supporting the increase in strain during the tablet deformation stage.

In addition, Figure 8d shows the relationship between the failure mode of the composites and the structural feature. Through the above analysis, we can come to a conclusion that the key to high toughness of the composites lies in the balance of two failure mechanisms: soft phase failure and tablet break. The soft phase failure helps to increase the fracture strain and improve the energy absorption, making the failure of the composite show pseudoplasticity, while the tablet break can improve the bearing capacity (i.e., the strength and stiffness) of the composite before its failure.

### 3.3. Simulation Results

Since the stress distribution of tablet is the concrete manifestation of the difference in mechanical behavior of the different bio-inspired composites, we perform numerical analysis for composites with different tablet waviness and wave number and illustrate the stress distribution in the tablet before fracture.

The tensile stress distribution in a tablet selected at the same position of different composites at the same overall tensile strain 0.65% (elastic deformation region) is displayed in Figure 9a. Firstly, an efficient stress transfer between the hard tablets and soft interface can be observed, thus the tensile stress in bricks is much higher than that in interfaces. It is evident that the stress is seen to concentrate in the center region of the tablets; specifically, the stress concentrates on the top of the wave in the wavy tablet surface. Therefore, as the waviness in tablets increases, the stress concentration area on the tablet surface gradually changes from a single, continuous one to multiple, discontinuous ones, but the total area of the stress concentration area is decreases. Notably, the number of stress concentration areas is the same as the wave number of the tablet when the waviness is not zero. In addition, the number of stress concentration area increases as the tablet wave number increases, however, the total area of the stress concentration area increases first and then decreases. Among all composites with different wave number but the same waviness, the ratio of the stress in the middle area of the tablet to that on the top of the wave in composite N8W20 is the smallest, which means that the tablet of composite N8W20 can withstand greater tensile deformation.

To further analyze the influences of tablet waviness and wave number on the stress distribution in composite while tensile, we compare the stress on the wave surface in the tablet, as shown in Figure 9b,c. It is clear that the stress increases as the tablet waviness increases, suggesting that more load is transferred to the tablets. Compared with the maximum stress at the top of the wave of tablet in composite N8W00, that of composite N8W30 increases by up to about nine times. The maximum stress of composite N8W30 quickly reaches the strength limit as the strain increases, leading to the earliest fracture of the tablet before the strong interface fails, which has been observed in experimental analysis of the composites. In contrast, the maximum stress at the stress concentration of composite N8W20 is second to that of composite N8W30, indicating that the stress increases at a slower rate, which is consistent with the delayed fracture of tablet of composite N8W20. The load transfer of composites with tablet waviness below 2 are relatively small, thereby the damage mainly occurs in the soft interface. Moreover, the maximum stress at the top of the wave of tablet decreases as the wave number increases. Compared with the maximum stress at the top of the wave of tablet in composite N6W20, that of composite N7W20, N8W20, N9W20 and N10W20 reduces by 3.9%, 3.1%, 9.0% and 14.1%, respectively. This indicates that the smaller wave number, the higher level of stress concentration, leads to easier crack growth as we have observed in the experimental analysis.

## 4. Conclusions

In this study, we proposed a hybrid brick-and-mortar-like structure by introducing the wavy suture structure of the woodpecker beak into the brick-and-mortar structure of the nacre. Compared with nacre-like composites with flat tablet, the strength, stiffness and toughness of the nacre-like composite with tablet of wave surface (N8W20) increase by up to 191.3%, 46.6% and 811.0%, respectively. This unusual combination of mechanical properties is an exciting result, especially the improved toughness achieved by the wave microstructure design, which is higher than that of the interlocking tablet design of the mineral bridge design [15,16,17,18,21]. Through an approach that integrates finite element simulations and experiments, we systematically investigated the influences of the tablet waviness and wave number on mechanical performance of the composites. Results show that the tablet waviness significantly affects the mechanical properties and failure patterns of the composite, while the tablet wave number only has a certain effect on the mechanical properties of the composite. By increasing the tablet waviness, the contact interface area of the tablet increases, thus providing a larger area for shear deformation during the hardening and promoting the transfer of load to the tablet. However, exaggerated tablet waviness may cause severe stress concentration, leading to localized brittle fracture of the tablet in composite. Three failure modes are observed in the tensile tests of bio-inspired nacreous composites: soft phase failure, soft phase failure coupled with tablet break and tablet break. Composites with the highest strength and toughness are contributed by the combination of soft phase failure and tablet break: providing adequate tablet sliding to delay fracture and enhancing the hardening to improve the energy absorption. The analysis of fracture path reveals that in the case of soft phase failure, the higher tablet waviness is conducive to the spread of cracks throughout the composite, thus promoting the overall deformation of the composite. In contrast, the change in geometric scale of wave-shaped design caused by the change of tablet wave number may also somewhat affect the load transfer to the tablet, and there is an optimal geometric scale of wave microstructure for the toughness of composite. The simulation verifies the stress distribution of the composites and proves that the high stress concentration area is confined to the peak of the wave in the center of the tablet. In this study, we conclude that the key to high strength and high toughness is to achieve the optimal balance between load transfer and stress concentration, transferring as much load to the tablet as possible while delaying the failure of the composite as much as possible. Thus, given the critical role of design revealed in this study, tuning fracture mode through design optimization in bio-inspired composites can improve mechanical properties of synthetic composites and can bolster the search for new functional advanced materials.

## Figures and Tables

**Figure 1 polymers-13-03681-f001:**
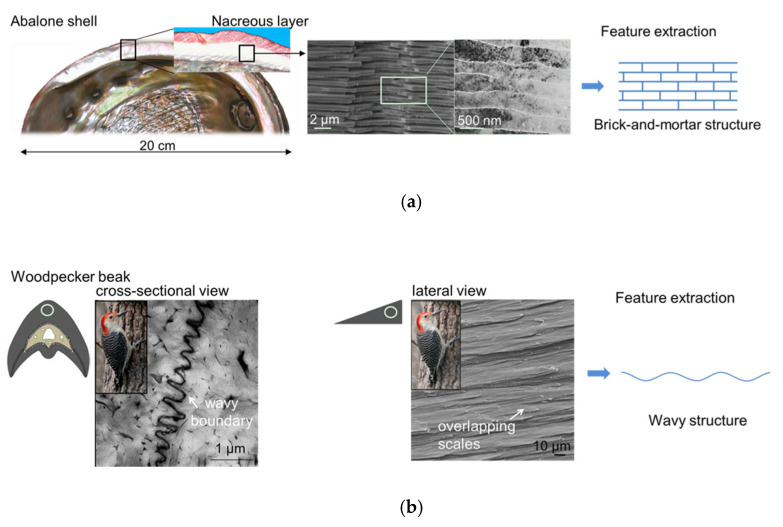
Biomimetic microstructures. (**a**) The typical microstructures of the nacre (adapted from Ref. [7]); (**b**) the typical microstructures of the woodpecker beak (adapted from Ref. [26]).

**Figure 2 polymers-13-03681-f002:**
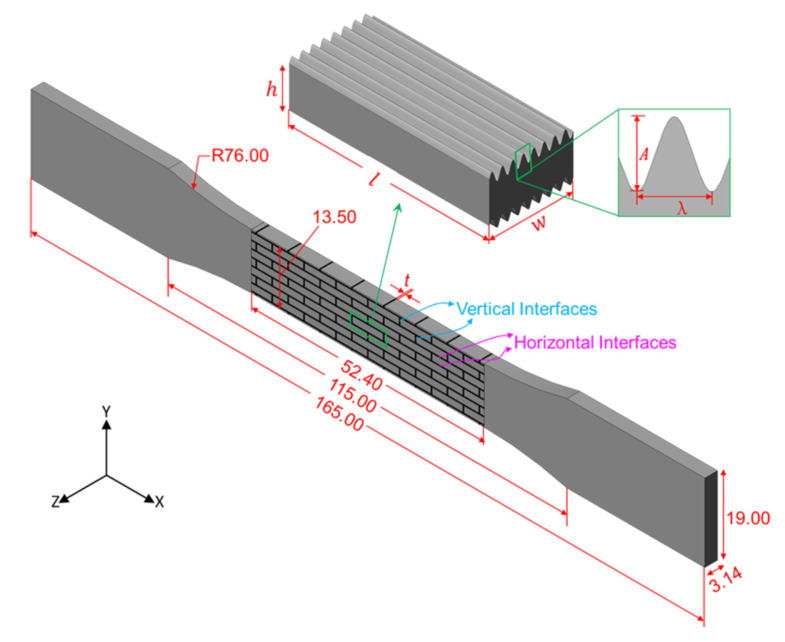
Geometric design of the hybrid bio-inspired structures in accordance with ASTMD368 standard and the unit-cell geometry of the samples with horizontal wavy interfaces.

**Figure 3 polymers-13-03681-f003:**
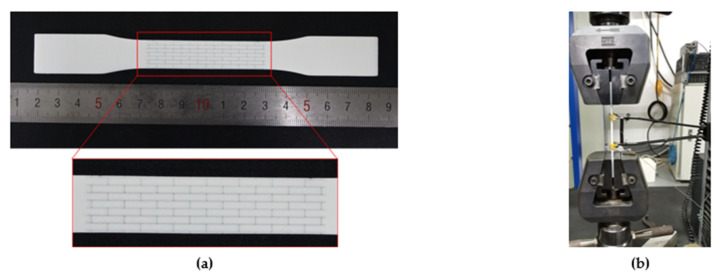
(**a**) A typical 3D-printed sample for tensile testing; (**b**) experimental setup for quasi-static tensile tests.

**Figure 4 polymers-13-03681-f004:**
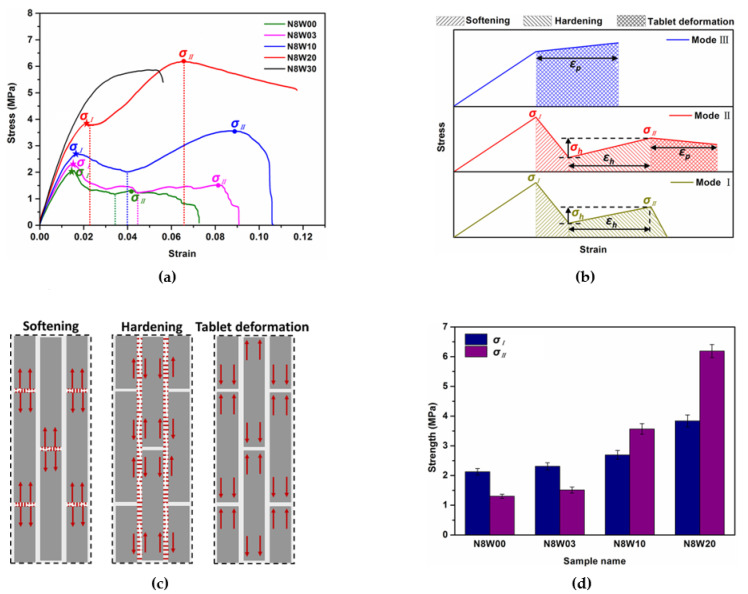
Tensile behaviors of composites with various tablet waviness. (**a**) Stress-strain curves of the composites with the tablet wave number of 8 and different waviness; (**b**) schematic of stress–strain curves for three different response modes; (**c**) schematic of failure patterns in the softening, hardening and tablet deformation stages; (**d**) comparison of the vertical short interface strength and horizontal interface strength of composites with different tablet waviness; (**e**) plot of increased stress and strain period of hardening stage; (**f**) plot of strain period of tablet deformation stage, energy absorption while hardening and tablet deformation of composites.

**Figure 5 polymers-13-03681-f005:**
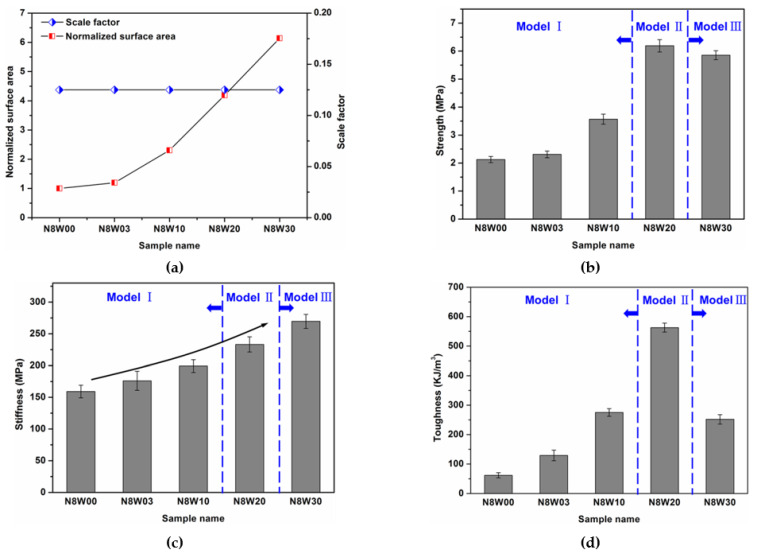
Effects of tablet waviness on the mechanical performance of composites. (**a**) The normalized surface area (normalized by the surface area of tablet in composite N8W00) and scale factor of composites; (**b**) strength; (**c**) stiffness; (**d**) toughness.

**Figure 6 polymers-13-03681-f006:**
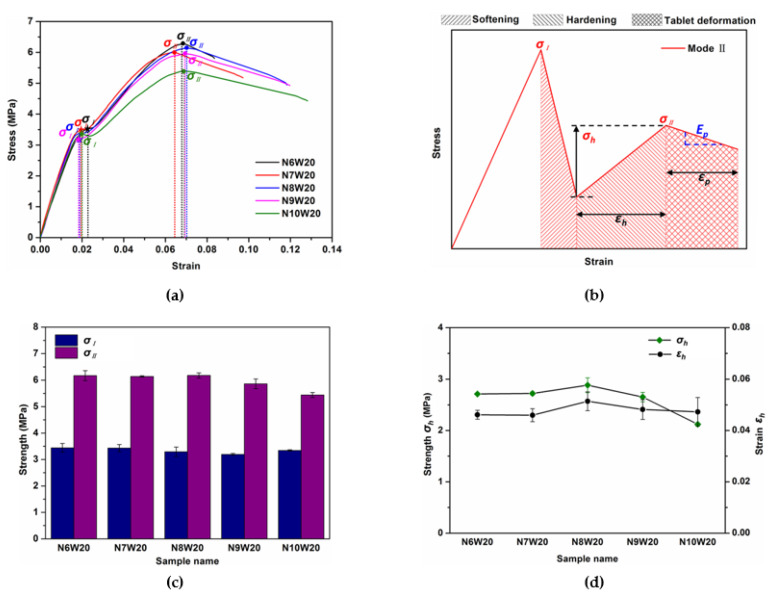
Tensile behaviors of composites with various tablet wave number. (**a**) Stress–strain curves of the composites with Table 2. and different wave number; (**b**) comparison of the vertical short interface strength and horizontal interface strength of composites with different tablet wave number; (**c**) plot of increased stress and strain period of hardening stage; (**d**) plot of strain period of tablet deformation stage, energy absorption while hardening and tablet deformation of composites; (**e**) plot of strain period of tablet deformation stage, energy absorption while hardening and tablet deformation of composites; (**f**) plot of the equivalent stiffness during the deformation stage.

**Figure 7 polymers-13-03681-f007:**
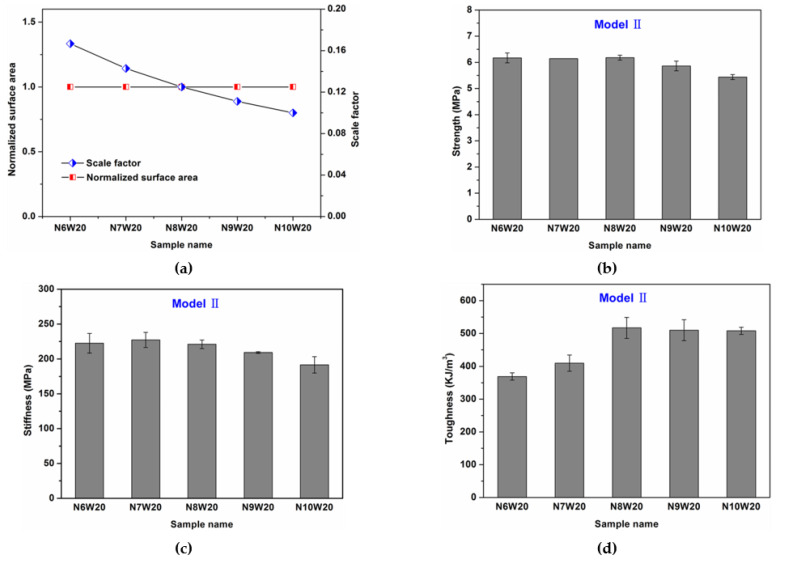
Effects of tablet wave number on the mechanical performance of composites. (**a**) The normalized surface area (normalized by the surface area of tablet in composite N6W20) and scale factor of composites; (**b**) strength; *(***c**) stiffness; (**d**) toughness. Schemes follow another format.

**Figure 8 polymers-13-03681-f008:**
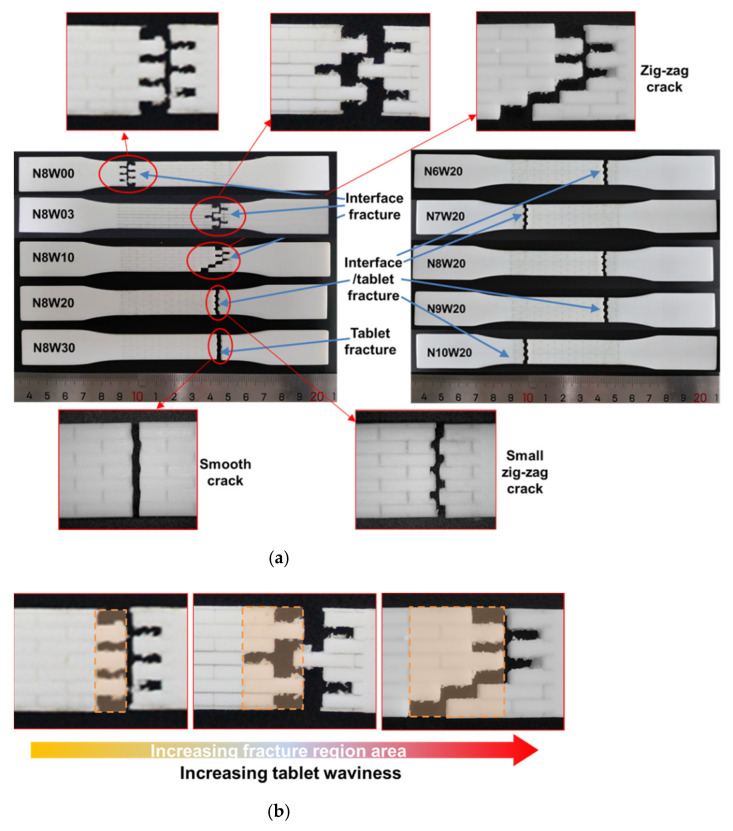
(**a**) Failure morphologies of the bio-inspired composites after tensile test; (**b**) schematics displaying that the increase in tablet waviness leads to the increase in of fracture region area; (**c**) quantification of fracture morphologies of composite with tablet waviness wv = 0, 0.3, 1, wave number of 8; (**d**) map of identified failure modes as they relate to the tablet waviness and wave number.

**Figure 9 polymers-13-03681-f009:**
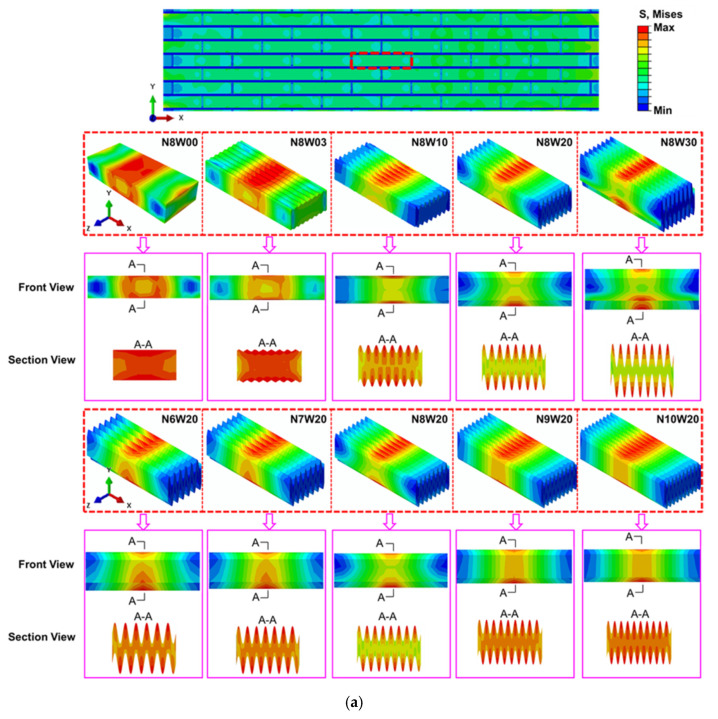
The simulation results of composites. (**a**) The stress distribution in tablets of composites with vari-ous tablet waviness at the same overall tensile strain based on specific table; (**b**) the comparison of the maximum stress at the stress concentration located in the center of tablet of composites with various tablet waviness; (**c**) The comparison of the maximum stress at the stress concentration lo-cated in the center of tablet of composites with various tablet wave number.

**Table 1 polymers-13-03681-t001:** Dimensions of the designs.

Design	*λ* (mm)	*A* (mm)	*h* (mm)	*l* (mm)	*t* (mm)	*w* (mm)	*f_v_* (%)
N8W00	0.3927	0	1.5	7.5	0.3	3.1415	78.994
N8W03	0.3927	0.1178	1.5	7.5	0.3	3.1415	78.993
N8W10	0.3927	0.3927	1.5	7.5	0.3	3.1415	78.992
N8W20	0.3927	0.7854	1.5	7.5	0.3	3.1415	79.001
N8W30	0.3927	1.1781	1.5	7.5	0.3	3.1415	79.876
N6W20	0.5236	1.0472	1.5	7.5	0.3	3.1415	79.635
N7W20	0.4488	0.8976	1.5	7.5	0.3	3.1415	79.373
N9W20	0.3491	0.6981	1.5	7.5	0.3	3.1415	79.071
N10W20	0.3141	0.6283	1.5	7.5	0.3	3.1415	78.990

**Table 2 polymers-13-03681-t002:** Mechanical properties of material used in numerical analysis.

Material	*E* (MPa)	*σ_b_* (MPa)	*v*
VeroWhite	1927	35	0.3
TangoPlus	3.5	1.2	0.4

## Data Availability

Data available on request due to privacy. The data presented in this study are available on request from the corresponding author. The data are not publicly available due to these data are also part of ongoing research.

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
