# Peer review of "Hybrid Bio-Inspired Structure Based on Nacre and Woodpecker Beak for Enhanced Mechanical Performance"

_polymers, 2021, doi:10.3390/polym13213681_

Round 1

Reviewer 1 Report

The manuscript by Ding et al titled 'Hybrid bio-inspired structure based on nacre and woodpecker beak for enhanced mechanical performance' is an interesting manuscript where by carefully designing the composite, materials' mechanical properties can be carefully tuned. The authors are suggested to proof-read the manuscript prior to publication.

Author Response

Dear Reviewer,

We would like to thank you for considering our manuscript entitled "Hybrid bio-inspired structure based on nacre and woodpecker beak for enhanced mechanical performance" (ID:polymers-1411051) for publication. The encourage comments and useful suggestions have been taken on board and the manuscript has been carefully proof-read to minimize typographical, grammatical, and bibliographical errors.

Yours sincerely,
Zhongqiu Ding (on behalf of all authors)

Reviewer 2 Report

This is an interesting study dealing with investigating a hybrid brick-and-mortar-like structure by introducing the wavy suture structure of the woodpecker beak into the brick-and-mortar structure of the nacre in terms of strength, stiffness and toughness of nacre-like composite with tablet of wave surface.

This manuscript presents a specific, easily identifiable advance in knowledge. It is applicable and useful to the profession. The title and abstract accurately describe the contents. The methodology is sufficiently explained. Language used in article is fluent. Classifications in tables and figures clearly represent experimental studies conducted before. Each figure and table is necessary to the understanding of the conclusions. The results are soundly interpreted and related to existing knowledge on the topic. The conclusions are sound and justified. They follow logically from data presented. All elements of the manuscript relate logically to the study's statement of purpose. Although the paper is valuable for publication in the “Polymers” journal, there are some minor comments that can improve the paper:

  • In the Introduction part, the authors need to narrow down the review towards the problem, highlighting the gaps in literature and ending with defining the problem statements and the objective of this research.
  • In the discussion part, the authors mainly described the results based on the graphs and figures, which looks like a report rather than a research work. It would be better if the results can be verified by the results of other papers in this field. Please compare your results with the results of other research works and try to confirm your results by adding some references in the discussion.
  • Please revise the beginning of Line 390.

Author Response

Dear Reviewer,
Thank you very much for reviewing our manuscript entitled "Hybrid bio-inspired structure based on nacre and woodpecker beak for enhanced mechanical performance" (ID:polymers-1411051). Your comments are very helpful for revising and improving our paper. We have studied the comments carefully and made revisions which we hope meet the approval. 
Please see the attachment for point-by-point response to the points rasied in your report.
Yours sincerely,
Zhongqiu Ding (on behalf of all authors)
